# In Vitro Evaluation of Anti-Rift Valley Fever Virus, Antioxidant and Anti-Inflammatory Activity of South African Medicinal Plant Extracts

**DOI:** 10.3390/v13020221

**Published:** 2021-01-31

**Authors:** Garland K. More, Raymond T. Makola, Gerhard Prinsloo

**Affiliations:** 1College of Agriculture and Environmental Sciences, University of South Africa, Private Bag X6, Florida, Johannesburg 1710, South Africa; prinsg@unisa.ac.za; 2Department of Biochemistry Microbiology and Biotechnology, School of Molecular and Life Science, University of Limpopo (Turfloop Campus) Sovenga, Polokwane 0727, South Africa; makolaraymond4@gmail.com; 3National institute of Communicable Diseases, Special Viral Pathogen/Arbovirus Unit, 1 Modderfontein Rd, Sandringham, Johannesburg 2192, South Africa

**Keywords:** rift valley fever virus, medicinal plants, antiviral activity, RTCA, H_2_DCF-DA

## Abstract

Rift valley fever virus (RVFV) is a mosquito-borne virus endemic to sub-Saharan African countries, and the first sporadic outbreaks outside Africa were reported in the Asia-Pacific region. There are no approved therapeutic agents available for RVFV; however, finding an effective antiviral agent against RVFV is important. This study aimed to evaluate the antiviral, antioxidant and anti-inflammatory activity of medicinal plant extracts. Twenty medicinal plants were screened for their anti-RVFV activity using the cytopathic effect (CPE) reduction method. The cytotoxicity assessment of the extracts was done before antiviral screening using the MTT assay. Antioxidant and reactive oxygen/nitrogen species’ (ROS/RNS) inhibitory activity by the extracts was investigated using non-cell-based and cell-based assays. Out of twenty plant extracts tested, eight showed significant potency against RVFV indicated by a decrease in tissue culture infectious dose (TCID_50_) < 10^5^. The cytotoxicity of extracts showed inhibitory concentrations values (IC_50_) > 200 µg/mL for most of the extracts. The antioxidant activity and anti-inflammatory results revealed that extracts scavenged free radicals exhibiting an IC_50_ range of 4.12–20.41 µg/mL and suppressed the production of pro-inflammatory mediators by 60–80% in Vero cells. This study demonstrated the ability of the extracts to lower RVFV viral load and their potency to reduce free radicals.

## 1. Introduction

Rift valley fever virus (RVFV) is an arthropod-borne virus, which belongs to the genus Phlebovirus under the family of Phenuiviridae (formerly: Bunyaviridae) that is characterized by a negative-single-stranded segmented tripartite RNA genome [1]. It is transmitted to humans through contact with infected unpasteurized milk, animal body fluids, aerosols exposure and/or bite from an infected mosquito. Human-to-human transmissions have not been reported. The main vectors of this virus are the *Culex* and *Aedes* mosquitoes, the latter is also considered as a reservoir [2]. The virus has emerged as one of the important etiological agents for an acute and self-limiting febrile illness and severe manifestations, which include hemorrhagic fever, retinitis, renal failure, encephalitis and miscarriages, in both humans and animals [3,4]. Since it affects livestock such as sheep, goats and camels, it has been reported to have a significant economic impact on meat and dairy industries with approximately 32 million USD loss due to livestock death in Kenya in the year 2007. An outbreak in the Arabian Peninsula accounted for an estimated 90 million USD loss in the year 2000 [2]. RVF is endemic to Sub-Saharan African countries and has caused numerous subsequent outbreaks in Africa, and this includes an outbreak in South Africa between 1950–1975 and 2010–2011, which was the first fatal record in humans. Other outbreaks were recorded in Zimbabwe in 1978, East Africa 1997–1998, Mauritania in 1998 and 2012, Kenya, Somalia and Tanzania 2006–2007, Niger in 2016, South Sudan in 2018 and Gambia in 2018 [5]. The RVFV was firstly recognized outside Africa in Saudi Arabia, Yemen in 2000–2001 and China in 2016 [6]. The most recent outbreak in Mayotte (France) was reported in November 2018–1 March 2019 with a high prevalence of cases in the rural areas of Centre-West and North of the island [7,8], and in August 2019, an outbreak was reported in the Central Africa Republic in the Bossembele health district [9].

Several arthropod-borne viruses (arboviruses), including the chikungunya virus (CHIKV), dengue virus (DENV), zika virus (ZIKV), rift valley fever virus (RVFV), yellow fever virus (YFV) and west Nile virus (WNV), are strong inducers of oxidative stress which then leads to various diseases such as liver damage, hemorrhagic fever and neuronal diseases [8,9]. Moreover, oxidative stress has been implicated to cause non-viral inflammatory diseases. Some of the diseases that are associated with excessive production of reactive oxygen/nitrogen species (ROS/RNS) in living cells include cancer, diabetes, cardiovascular disorders, neurodegenerative diseases, cataracts and osteoporosis [10,11,12,13]. Microbial infections have been demonstrated to trigger inflammatory responses via oxidative stress in infected cells and to develop therapeutic agents, it is important to understand the activation of innate immunity as a target for the management of infection-initiated oxidative stress. Oxidative-stress-related inflammatory diseases are intensifying and becoming a global health concern worldwide. Oxidative stress may be defined as an imbalance between pro-oxidant and antioxidant systems within a living cell [11]. An imbalance between the production of antioxidants such as superoxide dismutase, glutathione peroxidase and catalase as well as the non-enzymatic antioxidants such as glutathione and vitamins C, E and D and increased production of free radicals (reactive oxygen and nitrogen species) elicits an oxidative stress condition. Accumulation of ROS results in the oxidation of proteins, lipids and nucleic acids, DNA damage and disruption of cellular functionality [12,14]. 

During viral infections, ROS play an important role in cell signaling and contributes to the innate immune response to initiate pathogen killing. The activation of antioxidant defense pathways such as the nuclear factor E2-related factor 2 (Nrf2) and antioxidant-response element (ARE) play a crucial role in the pathogen-killing process [14,15]. Studies have shown that the upregulation of Nrf2 results in antioxidative cell defense against viral infections [16]. In contrast, secondary ROS signaling may result in the activation of transcription factors such as the nuclear factor kappa beta (NF-κB) that may lead to inflammatory responses and DNA damage in RVFV infections [17]. Studies have demonstrated that the downregulation of Nrf2 supports the viral genome replication and completion of the viral life cycle by upregulating the complexity of the NF-κB signaling pathway [14,15,18]. High levels of ROS have been shown to activate nitric oxide synthase 2 (iNOS 2) for RNS production. The presence of nitric oxide (NO) is beneficial in that it has antiviral properties, but in surplus, it can promote the pathogenesis of herpes simplex virus 1 (HSV-1) [19], human immunodeficiency virus (HIV) [20], influenza virus [21] and RVFV [22] by damaging cells in host tissues. Oxidatively, NO reacts with superoxide anion to form a highly reactive anion called peroxynitrite (ONOO^−^), which induces lipid peroxidation and nitro-tyrosination that disrupts the functioning of cells [14]. Furthermore, cellular enzymes such as superoxide dismutase (SOD), catalase (CAT) and glutathione peroxidase (GSHPx) are necessary for modulating free radicals into hydrogen peroxide (H_2_O_2_) and converting it into a less toxic element water (H_2_O) [23]. Therefore, suppression of pro-inflammatory mediators and blocking nuclear transcription of NF-κB has been proposed to be a good approach for the treatment of various oxidative-stress-induced diseases. 

Oxidative stress in viral infections has been shown to promote viral replication through free radical lipid oxidation, suppression of antioxidants and antiradical protection system in patients with neurotropic virus infections such as tick-borne encephalitis and herpes simplex virus [24]. Hence, the search for therapeutic agents that possess different mode of actions of antiviral potency has showed significant potentials. Our objectives in this study were to evaluate the antiviral activity as well as the antioxidant activities of selected active antiviral medicinal plant extracts in suppressing the DPPH, ABTS+ radicals and anti-inflammatory activity using the cell-based LPS-induced ROS/RNS assay.

There are no licensed vaccines or effective antiviral therapies available to treat RVF disease. Vaccines used in livestock have shown to present adverse effects and safety is a concern. Several molecules have been tested using cell culture and animal models, and they have shown antiviral potency. These molecules include two compounds with a 3,7-bis(dialkylamino)phenothiazine-5-ium skeleton, suramin an inhibitor of trypanosomiasis and onchocerciasis used in Africa, rapamycin [25], bortezomib [26], sorafenib [27], curcumin [28], ribavirin [29], favipiravir [30] and benzavir-2 [3]. However, there are no reports on the assessment of the latter molecules in clinical trials against RVFV [3,9]. Medicinal plants have been used as treatment or prevention against several human diseases and veterinary purposes for many years. Traditional medicines are generally cheaper, accessible or readily available and more culturally acceptable than “western” medicines [31]. For this reason, a polarized system of trade in medicines has been ongoing in South Africa, with herbal medicines offered by traditional healers and sold on traditional pharmacies, whilst “western” medicines are prescribed by physicians. To resolve this matter, there is currently an effort made to integrate these systems, and this has led to significant economic benefits and cultural heritage protection [32]. Although medicinal plants have been regarded as safe compared to synthetic drugs [33], there is an urgent need to scientifically study their efficacy. Plants have several secondary metabolites that display remarkable potential as sources of antiviral compounds. Compound classes such as alkaloids, flavonoids, tannins, terpenoids, saponins, steroids, glycosides and phenolic compounds are naturally associated with many biological activities including antiviral potency [34]. These bioactive compounds have prospects for use as anti-viral agents. The objective of this study was to evaluate the potential activity of the aqueous-methanolic crude extracts of twenty medicinal plants against RVFV infection using the neutralization CPE assay, which measures the inhibition of the cytopathic effects (CPE) on Vero cells. In this study, twenty medicinal plants with a broad-spectrum antiviral potency were selected pharmacologically, and interestingly, selected medicinal plants investigated for their antiviral activity against RVFV in this study have previously been reported to inhibit the infection of multiple viruses, including human immunodeficiency virus (HIV), herpes simplex virus type (HSV-1,2), African swine fever virus (ASFV), Newcastle disease virus (NDV), canine distemper virus (CDV), canine parainfluenza virus-2 (CPIV-2), feline herpesvirus-1 (FHV-1), poliovirus (PV-2), cytomegalovirus (CMV), lumpy skin disease virus (LSDV), coxsackie B virus (COX B-1), adenovirus 31 (AD-31), foot and mouth disease virus (FMDV), hepatitis B virus (HBV) and Epstein–Barr virus (EBV) (Table 1). To our knowledge, we report for the first time the inhibitory effect of South African medicinal plant extracts against the wild-type RVFV. The emergence of new viruses such as the currently tormenting Coronavirus (SARS-CoV-2), which are tough to control due to the mutative nature of their viral genomes and lack of therapeutic agents, highlights the importance of this type of research.

## 2. Materials and Methods

### 2.1. Cells, Viruses and Reagents

The RVFV AR 20368 strain was isolated during the 1974 RVF outbreak in South Africa. The virus was proliferated on a monolayer of Vero cells at a multiplicity of infection (MOI) of 0.2, followed by the harvesting of the monolayer at an extensive cytopathic effect (CPE) and centrifuged at 3000× *g* for 30 min. The virus supernatant was collected and stored at −80 °C [61]. The antiviral experiments were conducted at the national institute of communicable diseases (NICD) in the biosafety level-3 (BSL-3) laboratories as per RVFV classification [62]. African green monkey kidney (Vero) cells were purchased from Cellonex Separation Scientific SA (Pty) Ltd. (Roodepoort, South Africa), Johannesburg, SA and Dulbecco’s modified eagle’s medium, fetal bovine serum and penicillin-streptomycin, bought from Celtic Molecular Diagnostics SA (Pty) Ltd. (Cape Town, South Africa). Dimethyl sulfoxide (DMSO), were purchased from Sigma-Aldrich® Darmstadt, Germany. 

### 2.2. Plant Collection and Extraction 

Twenty different plant species known for their anti-viral properties against various viruses were selected based on their ethnobotanical uses for the treatment of viral infectious diseases and in vitro pharmacological antiviral activities obtained in the scientific literature. Plant leaves were collected from the Telperion Nature Reserve in Mpumalanga Province of South Africa. The leaves were air-dried at room temperature and pulverized to a fine powder using a grinding mill (IKA™ MF10 Mill, Munich, Germany). Fifty grams (50 g) of powdered plant material was extracted with 500 mL of 50% aqueous methanol, twice. Samples were filtered using a Buchi^®^ filtration system and concentrated under vacuum (EZ-2plus GeneVac™ evaporator, St. Louis, MO, USA). Concentrated crude extracts were each dissolved in 5% dimethylsulfoxide (DMSO) and filtered through a 0.2 µm sterile syringe filter to obtain a final stock solution of 10 mg/mL. 

### 2.3. Cytotoxicity Assays

The cytotoxicity assay was done using the 3-(4,5-dimethylthiazol-2-yl)-2,5-diphenyltetrazolium bromide (MTT) assay following a method by Mosmann, [63] with slight modifications. Vero cells were maintained in Dulbecco’s modified eagle’s medium (DMEM; Gibco) supplemented with 10% fetal bovine serum (FBS) and 1% penicillin/streptomycin in culture flasks and incubated at 37 °C and 5% CO2. When cells reach 85% confluency, cells were detached using 2% trypsin, and cell count was performed using an automated cell counter TC20™ (BIO-RAD). Vero cells were seeded at 1 × 10^4^ cells/100 µL overnight at 37 °C in a 5% CO_2_ incubator to allow cell attachment. After 24 h, treatments were administered with different concentrations of extracts ranging from (0 to 1000 mg/mL) and doxorubicin was used as a positive control with 5% dimethyl sulfoxide (DMSO) as a negative control. After a 24-h incubation, 20 µL of MTT solution (5 mg/mL in PBS) (Sigma-Aldrich, St. Louis, MO, USA; Merck, Darmstadt, Germany) was added followed by a 4-h incubation. DMSO (100 µL) was then added to dissolve the formazan crystals, and the optical density was measured at 540 nm using an ELISA microplate reader (VarioSkan Flash, Thermo Fisher Scientific, Vantaa, Finland). 

### 2.4. Antiviral Activity Assay

The antiviral activity of plant extracts against RVFV was evaluated using viral load quantification assay tissue culture infectious dose (TCID_50_) [64]. TCID_50_ is a cytopathic effect (CPE)-dependent assay that examines Vero cells monolayer integrity as a measure of viral load. Therefore, replicating viruses induces CPE; the inhibition of CPE is indicative of antiviral activity. Briefly, Vero cells were maintained in Dulbecco’s modified eagle’s medium (DMEM; Gibco) supplemented with 10% fetal bovine serum (FBS) and 1% penicillin/streptomycin in T75 culture flasks and incubated at 37 °C and 5% CO_2_. When cells reach 85% confluency, cells were detached using 2% trypsin, and cell count was performed using an automated cell counter (TC20™ BIO-RAD). RVFV at 1 × 10^4.8^ TCID_50_ (10-fold serially diluted) was suspended in DMEM and mixed with 100 µg/mL (non-toxic concentration) of each extract in sterile capped vials for 24 h at room temperature. The extract-virus suspension (100 µL) was incubated with 100 µL of cells for 7 days until CPE appeared. The CPE on cells was evaluated using a cell culture microscope by comparison with treated–untreated control cultures. The TCID_50_ values were calculated using the Spearman and Kärber algorithm method. Consequently, fewer CPE’s were considered as an indicator of antiviral activities of the extracts. Virus and cells only (positive control) and medium and cells only (negative control) were included in this experiment as controls. All experiments were conducted in triplicate.

### 2.5. Real-Time Cell Analyzer (RTCA)

The real-time cell analyzer system (RTCA) xCELLigence DP system (ACEA Biosciences, USA) is a current impedance-based assay that relates electrical impedance to cell index as determined by cell morphology, cell adhesion and cell number. The viability of RVFV infected Vero cells and the protective effects of infected cells by extracts were monitored by this system. Briefly, 100 μL of cell culture media was added to each well of the E-plate 16 (ACEA Biosciences Inc). The E-plate 16 was connected to the xCELLigence system to calibrate, optimize and obtain background impedance readings in the absence of cells within 2 min. After this reading, the media was aspirated, then 1 × 10^4^ cells/well were seeded in a 96 well plate and incubated for 1 h at 37 °C under 5% CO_2_, after which the E-plate 16 was connected onto the xCELLigence system located inside the incubator for continuous impedance recording for 24 h. The media was aspirated, and 100 μL of the virus with extract (100 μg/mL) was added, and the plates were allowed to stand for 2 h at 37 °C under 5% CO_2_ to allow adsorption of the suspension. Control wells containing cells only and virus-infected cells without extracts were included in this study. The E-plate 16 was re-connected onto the xCELLigence system and allowed to run for 170 h with cell index (CI) values measured every 15 min.

### 2.6. DPPH (2,2-Diphenyl-1-picrylhydrazyl) Radical-Scavenging Activity

The DPPH scavenging activity is based on the reduction in a free radical (DPPH) by a molecule that can donate protons. The assay was done using a method by Volsteedt [65] with minor modifications. Briefly, 20 μL of 1.0 mg/mL extracts was pipetted into 200 μL of methanol, added to the first top wells of a 96-well plate. The solutions were serially diluted onto the remaining wells of the 96-well plate that contained 110 μL of methanol. Ninety microliters of 0.1 mM methanolic DPPH was added to all the wells. The final concentrations of the extract ranged from 0.781 to 100 μg/mL. The plates were incubated for 30 min at 25 °C, and the absorbance was measured on the microplate reader (VarioSkan Flash, Thermo Fisher Scientific, Finland) at a wavelength of 517 nm. Ascorbic acid (Vitamin C) was used as the positive control at the same concentrations as the extracts. The DPPH-scavenging effects percentage was calculated using Equation (1):DPPH scavenging effects (%) = ((A0 − A1/A0))/A0 × 100(1)

A0 and A1 correspond to the absorbances at 517 nm of the DPPH in the absence and presence of antioxidants, respectively.

### 2.7. ABTS+ (2,2′-Azino-bis-3-ethylbenzthiazoline-6-sulphonic acid) Radical Scavenging Assay

The free radical scavenging activity of plant samples was determined by the ABTS+ radical cation decolorization assay. The experiment was conducted following the method as mentioned by Dzoyem et al. [66] with slight modification. ABTS+ cation radical was produced by the reaction between 10 mg of ABTS+ and 2 mg potassium persulfate in water. The solution was stored in the dark at room temperature for 12–16 h before use. ABTS+ solution (1 mL) was then diluted with 60 mL of methanol. Measurements were carried out in triplicate. Percent inhibition of absorbance at 734 nm [67] was calculated using Equation (2).
ABTS+·scavenging effect (%) = ((A0 − A1/A0))/A0 × 100(2)

A0 is the absorbance of ABTS+ radical and methanol; A1 is the absorbance of ABTS+ radical and sample extract, and the percentage inhibition of ABTS+ was represented using the effective concentration (EC50) value, which is the concentration of an antioxidant needed to reduce the radicals by 50%.

### 2.8. Lipopolysaccharides (LPS)-Induced Intracellular Reactive Oxygen/Nitrogen Species Measurement

The assessment of ROS was done using the Vero cells according to the method described by Wu et al., [33] with minor modifications. Cells were seeded at 1 × 10^4^ cells/well in a 96-well plate and incubated for 24 h at 37 °C in 5% CO_2_. After incubation, cells were treated with extract concentration (100 µg/mL) and the ROS stimulation at non-toxic doses of 1 µg/mL LPS [68], serving as the positive control. After 24 h incubation, 10 µM of the cell-permeant 2′,7′-dichlorodihydrofluorescein diacetate (H_2_DCFDA) was added for 30 min in the dark. The fluorescence was measured using a microplate reader (VarioSkan Flash, Thermo Fisher Scientific, Finland) at 485 and 535 nm excitation and emission, respectively. 

### 2.9. Nitrite Concentrations as a Measure of Reactive Nitrogen Species

Nitrite production was evaluated as an indicator of NO, measured in the supernatant of Vero cells. Nitrite was measured by adding 100 μL of Griess reagent (1% sulfanilamide and 0.1% naphthylenediamine in 5% phosphoric acid) to supernatants harvested from the ROS experiment. After a 10 min incubation, the optical density was measured at 550 nm in a microplate reader (VarioSkan Flash, Thermo Fisher Scientific, Finland).

### 2.10. Statistical Analyses

Experiments were done in triplicate or in quadruplets where necessary with three independent assay repeats, and results are given as mean ± standard deviation (SD). One-way analysis of variance (ANOVA) was used to determine the differences in means, and statistical processing of the data was performed using GraphPad Prism software (Version 8.0). Duncan’s multiple comparison t-test was used to determine significant differences between the means of treated and untreated groups. Differences were considered significant at * *p* ≤ 0.05; ** *p* ≤ 0.01, *** *p* ≤ 0.001.

## 3. Results

### 3.1. Cytotoxicity Results

Before evaluating the antiviral potential, the cytotoxicity of the extracts on the mammalian Vero cell line using the MTT assay was determined. Cells were treated with different concentrations of the extracts ranging from 8.0–1000 µg/mL. The results demonstrated dose-dependent toxicity with increased cell viability at lower concentrations and a decrease in cell viability with a gradual increase in concentrations (Appendix A). The lethal concentration that reduced cell viability by 50% (LC_50_) were recorded (Table 2) and the concentration that maintained 90% of Vero cell viability was defined as the maximum non-cytotoxic concentration (MNTC), which were tested for antiviral activity. All plant extracts displayed a varying degree of non-toxic behavior with an LC_50_ range of 82.0–400.6 µg/mL. The cytotoxic concentration that inhibited 50% of cell growth were all above 20 µg/mL, which denotes that plant extracts are safe at given LC_50_ values as compared to the positive control (Doxorubicin). Cell viability of less than 40% was observed at the highest extract concentration tested, and as the concentrations decreased, more viable cells were observed. 

### 3.2. Antiviral Results

The lethal concentrations that inhibited 50% of the cell viability (LC_50_) were recorded, and the concentrations that maintained 80–90% of Vero cell viability were defined as the maximal non-cytotoxic concentrations (MNTC), which were tested for antiviral activity. The antiviral activity was then performed using the MNTC (100 µg/mL) incubated with the virus for 24 h at room temperature. The extract-virus inoculum was added simultaneously to cells in their respective wells. In this study, tested concentration significantly decreased the viral titer by inhibiting the CPE induced by the virus infection in Vero cells. A threshold to classify activity was set to 10^5^ TCID_50_, and all extracts with an ability to lower the viral load to less than 10^5^ were considered highly active. As shown in Figure 1, eight extracts out of twenty significantly decreased the viral infection. The threshold was set at 10^5^ TCID_50_, since the extracts lower viral load by 2 logs compared to the virus control. 

### 3.3. Real-Time Cell Analyzer (RTCA)

According to records from the RTCA, the cell index (CI) values increased slightly within 24 h prior to treatment and post-treatment. The CI values post-treatment showed a continuous stable increase between 24–90 h; this phase post-treatment can best correlate to viral incubation and adaptation in cells. Only after 90 h, a rapid exponential growth phase (proliferative phase) was observed between 90 and 120 h, with control viral infected cells having decreased CI values after 110 h, which is indicative of viral-induced cell death (Figure 2). Extracts treated cells then moved to the stationary phase in 120–140 h and a steady drop in CI values was experienced, which reflects morphological changes and loss of viability. The results correlate to the MTT cytotoxicity and antiviral assay as *E. croceum* and *A. afra* were the best viral-induced CPE inhibitors followed by *A. digitata*, *E. transvaalese*, *E. natalensis*, *H. aureonitens* and *S. frutescens*. 

### 3.4. DPPH Radical-Scavenging Activity 

The principle of the 2,2-diphenyl-1-picrylhydrazyl (DPPH) scavenging activity assay is based on measuring the color change in the radical DPPH in the presence of an antioxidant. The EC_50_, which can be defined as the concentration of an antioxidant needed to reduce the absorbance of DPPH by 50% from the initial absorbance measured after 30 min, was calculated, and the goodness of fit for the graphs was observed through non-linear regression where the value of the R-squared (*R*^2^) ≥ 0.900. Eight extracts demonstrated a noteworthy dose-dependent response to DPPH scavenging activity. It was observed that among the eight extracts, five extracts exhibited EC_50_ values < 10 μg/mL with *A. digitata* (EC_50_ = 4.64 μg/mL) being the most potent extract, followed by *E. natalensis* (EC_50_ = 5.30 μg/mL), while *S. frutescens* had the least DPPH scavenging activity. These results were comparable to the positive control, ascorbic acid (EC_50_ = 2.50 µg/mL).

### 3.5. ABTS+ Radical Scavenging Acvtiity

To evaluate the 2,2′-Azino-bis(3-ethylbenzothiazoline-6-sulfonic acid) diammonium salt (ABTS) scavenging activity of eight selected plants, radicals were formulated through the reaction of ABTS+ with potassium per-sulphate (K_2_S_2_O_8_) in sterile distilled water. This blue-green solution turns light-green when it is reduced by hydrogen-donating antioxidants, and the reduction may be quantified spectrophotometrically. In this study, a photometric measurement of the reaction between extracts and ABTS+ radical was quantified after 30 min incubation at room temperature and EC_50_ values were calculated from a scatter plot using a non-linear regression line, which showed a good fit coefficient of *R*^2^ ≥ 0.900. The results of eight extracts demonstrated the reduction in ABTS radical, and the EC_50_ values were recorded as shown in Table 2. Similar to DPPH scavenging results, out of eight extracts, *E. croceum*, *E. natalensis* and *A. digitata*, in respective order, showed significantly higher ABTS+ reducing power with EC_50_ values < 10 μg/mL, while *H. aureonitens*, *E. transvaalense*, *E. elephantina* and *S. frutescens* had EC_50_ values > 10 μg/mL. It is worth noting that the observed high ABTS scavenging activity of the extracts has comparable significant EC_50_ values to the positive control, ascorbic acid, which exhibited good ABTS reducing power with an EC_50_ value of 2.30 μg/mL. 

### 3.6. Measurement of LPS-Induced Intracellular ROS

As plant extracts have shown potent radical scavenging activities, it is worth evaluating the protective action of the extracts against oxidative stress. This can be done by assessing the effects of the extracts on the intracellular ROS content in Vero cells by inducing exogenous oxidative stress using Lipopolysaccharide (LPS), which is an endotoxin from the Gram-negative bacterium *Escherichia coli*. ROS production was detected using the cell-permeant probe, 2′,7′-dichlorodihydrofluorescein diacetate (H_2_DCFDA), which is non-fluorescent, but it is converted to the highly fluorescent 2′,7′-dichlorofluorescein (DCF) when acetate groups are cleaved by intracellular esterase’s [69]. The results demonstrate that ROS levels significantly increased to approximately 95% after exposure to LPS alone compared to the control group (untreated cells). However, pre-treatment with extracts (100 µg/mL) considerably reduced the LPS-induced ROS levels, as shown in Figure 3A. *Adansonia digitata* extract reduced the ROS levels dramatically with 75% reduction, followed by *E. croceum, E. transvaalense*, *E. elephantina* and *E. natalensis* with more than a 60% reduction and *A. afra*, *H. aureonitens* and *S. frutescens* exhibiting less than a 50% reduction. 

### 3.7. Measurement of LPS-Induced Intracellular RNS

The RNS experiment was designed to evaluate the inhibition of nitric oxide (NO) production in Vero cells pre-treated with the extracts for 24 h. As shown in Figure 3B, upon stimulation with LPS (1 μg/mL), an escalated amount of NO was produced when compared to unstimulated cells. Treatment with extracts (100 μg/mL) significantly reduced the levels of LPS-induced NO. Though *E. elephantina* extract showed moderate DPPH and ABTS+ radical scavenging activity, in this case, it exerted the highest suppression activity of NO with inhibition >80%. Whereas, *E. croceum*, *E. transvaalense* and *E. natalensis* extracts exhibited substantial NO inhibitory activity of approximately 80, 78 and 75%, respectively. Similar to the LPS-induced ROS inhibitory activity, *A. afra*, *H. aureonitens* and *S. frutescens* showed reasonable attenuation of NO production with a 50% inhibitory effect. Our results suggest that the inhibitory activity of reactive species by different plant extracts is dependent on diverse metabolites within the extracts, which exerts various modes of actions. 

## 4. Discussion

In recent years, the emergence of viral infectious diseases is one of the principal public health challenges, and the lack of licensed therapeutic agents make it difficult to contain infections. Faced with the advent of viral outbreaks, failure to contain the spread of the Coronavirus (COVID-19) outbreak is proof that more research should be done to develop new antiviral agents. However, investigation of medicinal plants as potential alternative therapeutic agents is necessary, especially in developing countries. There has been an enormous interest in developing and repurposing of drugs from natural products, with the recent outbreak of Coronavirus prompting researchers to investigate the use of antimalarial treatments such as artemisinin and its derivatives from the *Artemisia annua* plant. 

In our study, we evaluated the anti-RVFV activity of 20 medicinal plants used in traditional medicine and pharmacologically proven potency against viruses. Vero cells were the preferred cell line to work with in this study, as they are regarded as the most susceptible cell line for virus-mediated cell death and virus propagation. They have been generally utilized in toxicology, virology and pharmacology research, as well as in the production of vaccines and diagnostic reagents [70,71]. Furthermore, compared to other normal mammalian cells, e.g., leukocytes and fibroblasts, it does not secrete interferon (INF)-α or -β when infected by viruses [72]. The aqueous-methanolic leaf extracts exhibited relative non-toxic effects on Vero cells tested at concentration range of 8.0–1000 µg/mL, with only *A. afra* and *R. communis* showing values <200 µg/mL. *Ricinus communis* exhibited a significantly low LC_50_ of 82.0 µg/mL, which was expected, since the plant is known to be toxic to humans and animals containing the highly toxic constituents of this plant namely ricin and ricinine [22,23]. In addition, *E. croceum* was expected to be among the extracts with a low LC_50_ value, since its toxicity has been reported. Digitoxigenin-glucoside from *E. croceum* extracts was evaluated for its cytotoxic effects on Vero cells using the colorimetric (MTT) assay. Results from this study showed that digitoxigenin-glucoside possesses cytotoxic effects with a recorded 20% cell viability at a concentration of 25 μg/mL [46]. Other *E. croceum* constituents 20-hydroxy-20-epitingenone and tingenone tested against HeLa, MCF-7 and SNO cell lines exhibited toxicity with reported IC_50_ values ranging between 2.5 to 0.4 μM [73]. *Aloe ferox*, well renowned for its healing properties and commercialized for its use in pharmaceuticals, cosmeceuticals and nutraceuticals, was the second extract that showed relatively non-toxic behavior with an LC_50_ of 330.3 µg/mL. Celestino et al. [74] investigated the toxicity of *A. ferox* resin extract at doses of 50, 100 and 200 mg/kg; the results showed that *A. ferox* resin induced a significant increase in gastrointestinal motility with an ED_50_ of 19.1 mg/kg, and they concluded that *A. ferox* is laxative and non-toxic. 

The extracts investigated in this study significantly reduced the RVFV infection in Vero cells. The *E. croceum* extract significantly decreased the number of TCID_50_ up to 10^2^ log of the RVFV infectivity followed by *A. afra* and *A. digitata* with a reduction of up to 10^4^ log. An average potency compared to *E. croceum* was observed in *E. transvaalese*, *E. natalensis*, *H. aureonitens* and *S. frutescens*, which lowered the RVFV viral load when the virions were treated with 100 µg/mL of the extracts, which had a TCID_50_ activity range of 10^4^–10^5^. Furthermore, the real-time monitoring of RVFV infectivity in Vero cells showed inhibitory effects of extracts on RVFV-infected cells. An increase in cell index (CI) values was observed in pre- and post-treatment, with a lag phase from 24 h until 90 h. The lag phase may be due to a low growth rate of Vero cells inoculated at minimum cell concentration to accommodate the 7 days antiviral duration according to Vero cells kinetics [75]. From 90 h, the culture entered an exponential phase and reached the optimum cell viability at 110–140 h in *A. digitata*, *E. elephantina*, *E. transvaalese, E. natalensis, H. aureonitens* and *S. frutescens* treated cells, while *E. croceum* and *A. afra* treated cells showed maximum cell viability at 140–160 h. The results convincingly demonstrate that extracts tested exhibit antiviral potency against the RVFV, although it would be important to assess the mechanism(s) of action of these extracts.

Generally, it is well known that pre-incubation of the virus with the extracts changes the morphology of the viral particles. This, however, may lead to antiviral properties in different mechanisms including inhibition of viral attachment and entry to host cells. Therefore, our results suggest that medicinal plants tested, and their constituents may interfere with the viral replication through various mechanisms including the binding of virion to the cellular receptors, therefore disabling the viral entry to the host cells [76]. The relatively high antiviral activity shown by *A. afra* and *E. croceum* extracts could be attributed to the presence of constituents such as chlorogenic acids especially 4,5-dicaffeolyquinic acid [39], epigallocatechin-gallate [44,77] and gallic acid [78]. 

Eight extracts with the best antiviral activity were screened for their free radicals scavenging potential. These extracts significantly inhibited the DPPH and ABTS+ free radicals in a dose-dependent manner (Appendix A). Among eight tested extracts against the DPPH radical, five extracts exhibited EC_50_ values < 10 μg/mL with *A. digitata* (EC_50_ = 4.64 μg/mL) and *E. natalensis* (EC_50_ = 5.30 μg/mL) showing the best activity. Irondi et al. [79] reported that *A. digitata* possesses a strong DPPH scavenging potential with scavenging concentration (SC_50_) of 0.23 ± 0.01 mg/mL, which is moderate in comparison with the results of this study. We employed the DPPH and ABTS+ radical scavenging methods to investigate the antioxidant activity of plant extracts that showed antiviral potency. The DPPH method is based on the reduction in a stable electron/hydrogen acceptor, which is reduced to DPPH [80], while a cation ABTS+ reacts with antioxidants by an electron transfer mechanism [81]. In our study, evaluation of the DPPH and ABTS+ scavenging activity showed approximately similar values, which suggests that these two methods are positively correlated. Lipopolysaccharides (LPS) is a major component of the microbial cell membrane, and it plays a crucial role in tissue and organ injury in both humans and animals [82]. Furthermore, LPS induces the secretion of inflammatory cytokine and mediators through the elevated formation of ROS/RNS such as superoxide radical (O_2_•^−^), lipid peroxides and nitric oxide, which cause oxidative stress. The reduction in LPS-induced ROS and RNS showed essentially identical results in our study. Although different radical scavenging methods were used, the high correlation in antioxidant methods and anti-inflammatory methods suggest that certain specific metabolite is involved in quenching the radicals by stimulating pathways that target different radicals but may be using different mechanisms, this agrees with the work of other studies [81,82,83].

It was observed that the DPPH scavenging activity of the extracts seemed to correspond directly to the ABTS+ scavenging activity with *E. croceum*, *E. natalensis* and *A. digitata* exhibiting the best activity (EC_50_ = 4.12, 5.00, 5.04 μg/mL), respectively. Previous evaluation of DPPH and ABTS+ radical scavenging activities of *E. croceum* acetone leaf extract yielded significant results with IC_50_ values of 7.7 and 3.1 μg/mL, respectively [66], comparing well the results obtained in this study. Maroyi [84]; Odeyemi and Afolayan [85], however, reported much lower DPPH and ABTS+ scavenging activities of *E. croceum* leaf and bark acetone extracts. In their study, they observed IC_50_ values 0.09 and 0.1 mg/mL for the leaf extracts in ABTS and DPPH assay, while the bark extract showed activities with IC_50_ of 0.2 and 0.07 mg/mL ABTS and DPPH assay, respectively. Additionally, extracts also showed the reduction in LPS-induced ROS and RNS with high activity >60% by *E. croceum*, *E. natalensis* and *A. digitata* in both ROS and RNS assay. The results obtained in the study correlate well with the good radical scavenging activities. Suppression of ROS may indicate why the extracts exhibited significantly less toxicity on Vero cells, since cellular oxidative stress and cell viability assays can be indicators of elevated ROS production, which may result in cell death [86]. 

Plants investigated in this study strongly inhibited the DPPH and ABTS+ radicals and reduced the production of LPS-induced reactive oxygen species. Mechanisms of antioxidants and oxidants may modulate excessive ROS/RNS production and oxidative stress in living cells, and this would be a promising strategy for the treatment of viral-induced inflammatory disorders. Several studies have demonstrated the ability of natural products in activating cytoprotective pathways especially the upregulation of Nrf2 and ARE in promoting the suppression of viral replication in vitro and in vivo. Epigallocatechin-3-O-gallate (EGCG) from tea exhibits anti-HIV activities. Possible mechanisms of action were outlined through the modulation of Tat-induced LTR transactivation via the elevation of nuclear Nrf2 and decreasing the NF-κB levels by EGCG [87]. This study is in agreement with another study where the anti-influenza virus and suppression of virus-induced oxidative stress in vitro resulted in inhibition of viral replication by curcumin. This inhibitory activity of influenza virus by curcumin was through the upregulation of the Nrf2 signal and increased interferon-beta (IFN-β) production [88]. On the other hand, mononuclear phagocytic cells infected with DENV activated the Nrf2 pathway for cytoprotection; however, when Nrf2 inhibitor all-trans retinoic acid (ATRA) was added, a significant decrease in DENV-induced Nrf2 activity was observed, and this resulted in inhibited c-type lectin domain family 5, member A (CLEC5A) and tumor necrosis factor-(TNF-)α expressions, which led to an increase in the survival rate in suckling mice during DENV infection [89]. The anti-hepatitis B virus (HBV) effect of 3,4-*O*-Dicaffeoylquinic acid (DCQA) reduces the stability of the HBV core protein, which blocks the refill of nuclear HBV cDNA. Furthermore, the hepatoprotective effect of this compound may be mediated through its antioxidative/anti-inflammatory [90]. Another study presented possible mechanism of action of antioxidant in conjunction with the antiviral activity of quinic acid, a component of the CQA compounds, with potent anti-HIV activity. This showed that quinic acid binds to the HIV-RT enzyme, thereby inhibiting multiplication in the cell [91]. Compounds such as Spiroketalenol ether derivatives isolated from rhizome extract of *Tanacetum vulgare*, which function as cell entry inhibitors had been reported to block virus entry and also arrest the synthesis of HSV-1 gC and HSV-2 gG glycoproteins. Samarangenin B from roots of *Limonium sinense* exhibited inhibition of HSV-1 α gene expression. *Artocarpus lakoocha* with oxyresveratrol was found to inhibit the early and late phase of viral replication of HSV-1 and HSV- 2, respectively [92]. Other studies have demonstrated that natural antioxidants potentiate antiviral activity, since they suppress the ROS production in infected cells and downregulates transcription factor NF-κB. Fedoreyev et al. [24] showed the antioxidant and antiviral activities of echinochrome (Ech) formulation composed of tocopherol (Toc) and ascorbic acid (Asc) against tick-borne encephalitis virus (TBEV) and herpes simplex virus type 1 (HSV-1). Echinochrome formulations Ech, Ech + Asc + Toc and Asc + Toc decreased the ROS production by 61, 68 and 50%, respectively, in Vero cells. Antioxidant activity was investigated using the linetol peroxidation method and echinochrome formulations exhibited antioxidant potency with the formulation of Ech + Asc + Toc showing strong antioxidant activity. Furthermore, inhibitory concentrations (IC_50_) values of echinochrome formulations were 12.6, 21.8, 1304 μg/mL for Ech + Asc + Toc; Ech; Asc + Toc against TBEV, respectively, while IC_50_ values of formulations against HSV-1 were 11.2, 18.8 and 885 μg/mL, sequentially. This has demonstrated that Ech inhibits viral particles and indirectly enhance antioxidant defense mechanisms. Another study investigated the antiviral and antioxidant activity of a hydroalcoholic extract from *Humulus lupulus*. The DPPH and the ABTS+ assays were used to demonstrate the antioxidant activity, the viral titration and mRNA quantification to investigate the antiviral activity on influenza A virus strains: human A/Puerto Rico/8/34 H1N1 (PR8), A/NWS/33 H1N1 (NWS) and pandemic A/California/04/09 H1N1 (pH1N1) strains or avian Parrot/Ulster/73 H7N1 (ULSTER). This study showed that *H. lupulus* significantly reduced pH1N1, PR8 and ULSTER titer by 75, 44 and 29%, respectively, with IC_50_ values of 574.1 and 311.1 μg/mL for DPPH and ABTS+ scavenging activity [93].

In summary, increased production of ROS/RNS plays an important role in neutralizing many viral-induced inflammatory responses. However, ROS/RNS modulate the tolerance of cells to viral replication and regulate host inflammatory and immune responses, thereby resulting in oxidative damage to both host tissue and progeny virus. Even though there are debates about the role played by elevated ROS/RNS levels in cell signaling, studies have shown that increased levels ROS/RNS lead to a compromised immune system with severe oxidative injuries and viral disease progression. To curb the viral survival, the presence of antioxidants is crucial, as they activate Nrf2 and ARE defense pathways that lead to antioxidant defense system against virus-induced inflammation. In our study, intercellular interaction between the extracts and virus resulted in the reduction in the viral proliferation, possibly by oxidizing cell membrane proteins that are responsible for attachment/entry of the virus to mammalian cells. Therefore, bridging this attachment capacity may lead to failure of viral propagation. Moreover, intracellular reduction in ROS/RNS production may lead to the activation of the antioxidant defensive pathway and suppression of the viral replication by downregulating the NF-κB pathway that promotes oxidative stress and viral progression. Our preliminary results showed that plant extracts modulated ROS/RNS production in Vero cells, and they affect the morphology of the viral particles. Further experiments are warranted to investigate how the viral infection proceeds on Vero cells after treatment with the potent ROS/RNS plant extracts, since our study investigated the virus–extract interactions on Vero cells. Moreover, these experiments will validate whether the antioxidant and ROS/RNS activity shown by the plant extracts contributes to the antiviral activity observed. In addition, we plan to investigate mechanism(s) of action and metabolomics analysis of the extracts to better understand chemical constituents that play a role in antioxidant and antiviral activity.

## 5. Conclusions

RVFV infections have been associated with hemorrhagic fever and liver damage. Our study has demonstrated that the medicinal plants tested have the viral-induced cytopathic cellular protective ability and they act as viral inhibitors. In addition to conventional in vitro cell monitoring assays, such as the MTT colorimetric method, RTCA has proven to be an innovative technology to obtain information on the behavior, dissemination and the well-being of cells in the assessment of antiviral activity. Mechanisms of antioxidants and oxidants may modulate excessive ROS/RNS production and oxidative stress in living cells, and this would be a promising strategy for the treatment of viral-induced inflammatory disorders. The best antiviral activity was obtained with *E. croceum*, followed by *A. afra* and *A. digitata*. These plants, in addition to *E. natalensis*, showed consistently the best activity in the DPPH, ABTS and LPS-induced ROS and RNS assays and will be subject to further investigation. Prospectively, we aim at understanding the mechanisms that contribute to viral inhibition by investigating the effects of plant extracts against viral DNA replication, viral attachment mediators to cells and immunological activation or suppression of inflammatory mediators such as NF-κB and TNFα in cells. In addition, liquid chromatography mass spectrometry (LC-MS), nuclear magnetic resonance (NMR) metabolomics coupled with multivariate statistical analysis will be employed to identify metabolites responsible for the anti-RVFV activity in active plant extracts.

## Figures and Tables

**Figure 1 viruses-13-00221-f001:**
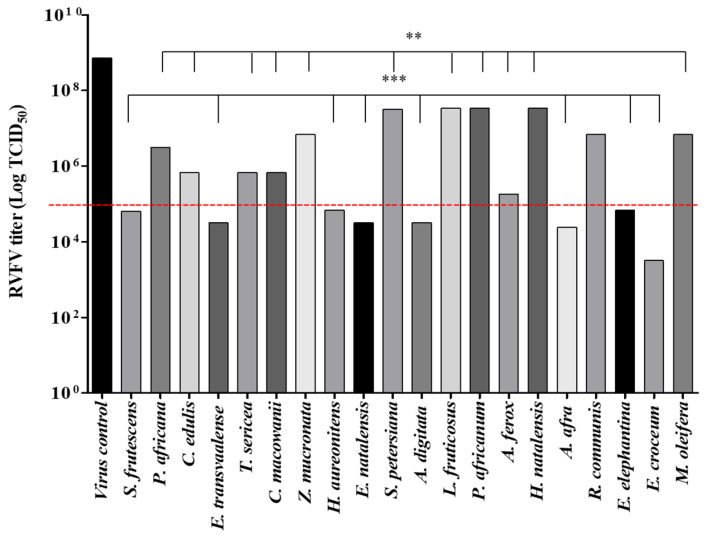
Effects of plant extracts on the RVFV tissue culture infectious dose (TCID_50_). The red dotted horizontal line shows the limit of viral susceptibility/resistance to extracts. The data represent the means ± SD from three independent experiments. A one-way ANOVA and Duncan’s test for multiple comparisons were used for the statistical analysis (** *p* < 0.01; *** *p* < 0.001).

**Figure 2 viruses-13-00221-f002:**
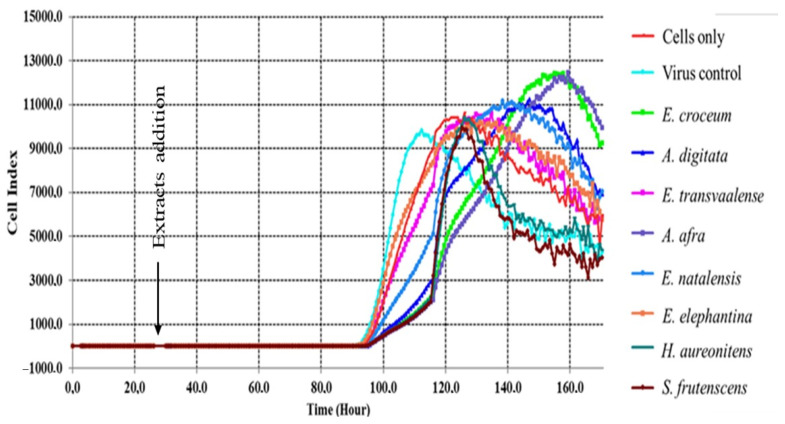
Real-time monitoring of the antiviral effects and cell growth in response to viral infection and/or extract treatment using the xCELLigence RTCA analyzer. Colored curves represent the Vero cells (control), viral-infected cells and extract (100 µg/mL) treated viral-infected Vero cells.

**Figure 3 viruses-13-00221-f003:**
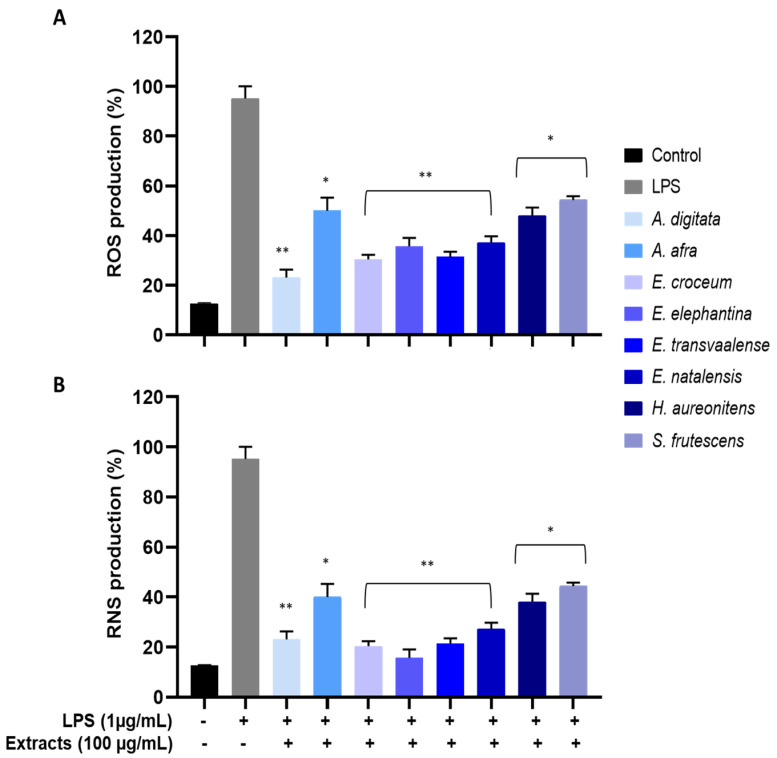
Intracellular reactive oxygen species (ROS) (**A**) and reactive nitrogen species (RNS) (**B**) scavenging activities of eight extracts against LPS-induced oxidative stress in Vero cells. Statistical analysis: one-way ANOVA with Duncan multiple range test for means ± SD (*) *p* < 0.05, (**) *p* < 0.01 was considered to indicate statistically significant difference compared to the LPS-treated group.

**Table 1 viruses-13-00221-t001:** Plants with reported pharmacological anti-viral activity, their family names and antiviral activities.

Plant Names	Family	Antiviral Activities	References
*Adansonia digitata*	Malvaceae	HSV-1, NDV, HSV-1; ASFV, HIV1 RT, HIV-FRET, PR	[33,34,35,36,37,38]
*Artemisia afra*	Asteraceae	HIV-1/2	[37,39]
*Aloe ferox*	Asphodelaceae	HSV-1	[40]
*Carissa edulis*	Apocynaceae	HSV-1, CDV, CPIV, FHV, LSDV, PV-2, CMV	[40,41,42,43,44]
*Crinum macowanii*	Amaryllidaceae	HIV-1 RT, PR	[45]
*Elaeodendron croceum*	Celastraceae	HIV-CB	[46]
*Elaeodendron transvaalense*	Celastraceae	HIV-1 a-Glucosidase, RT, CB, NF-kB, Tat, IN	[45,46,47]
*Elephantorrhiza elephantina*	Fabaceae	HIV-RT	[48]
*Euclea natalensis*	Ebenaceae	HIV-1 RT, HSV-1	[49]
*Helichrysum aureonitens*	Asteraceae	HSV-1, Cox B-1, Ad31 reovirus	[50]
*Heteropyxis natalensis*	Heteropyxidaceae	HIV-1 RT	[51]
*Lobostemon fruticosus*	Boraginaceae	HIV-1	[52]
*Moringa oleifera*	Moringaceae	HSV1, HIV-1 RT, FMDV, HBV, EBV	[50,51,52,53,54,55]
*Peltophorum africanum*	Fabaceae	HIV1-RT	[56,46]
*Prunus africana*	Rosaceae	CMV	[47]
*Ricinus communis*	Euphorbiaceae	HIV1- RT, RNase H, HIV-1 IN	[45,46,57]
*Senna petersiana*	Fabaceae	HIV1-RT	[58]
*Sutherlandia frutescens*	Fabaceae	HIV1 RT, IN, RNase H	[45,46,59]
*Terminalia sericea*	Combretaceae	HIV1 RT, HIV-1 RNA-dependent-DNA polymerase (RDDP)	[47,46]
*Ziziphus mucronata*	Rhamnaceae	HIV-1 RT, RNase H	[60]

Human immune deficiency virus (HIV-1,2), herpes simplex virus type (HSV-1,2), African swine fever virus (ASFV), Newcastle disease virus (NDV), canine distemper virus (CDV), canine parainfluenza virus-2 (CPIV-2), feline herpesvirus-1 (FHV-1), poliovirus (PV-2), cytomegalovirus (CMV), lumpy skin disease virus (LSDV), coxsackie B virus (COX B-1), adenovirus 31 (AD-31), foot and mouth disease virus (FMDV), hepatitis B virus (HBV), Epstein–Barr virus (EBV).

**Table 2 viruses-13-00221-t002:** Cytotoxicity effects of extracts from 20 plants with antiviral activity. Lethal concentration (LC_50_), DPPH EC_50_ and ABTS+ EC_50_ values in µg/mL.

Plant Names	LC_50_ (µg/mL)	DPPH EC_50_ (µg/mL)	ABTS EC_50_ (µg/mL)
*Adansonia digitata*	291.5	**4.64**	**5.04**
*Artemisia afra*	151.8	20.41	16.39
*Aloe ferox*	330.3	NT	NT
*Carissa edulis*	400.6	NT	NT
*Crinum macowanii*	389.4	NT	NT
*Elaeodendron croceum*	394.4	**6.00**	**4.12**
*Elaeodendron ransvaalense*	336.9	11.64	15.00
*Elephantorrhiza elephantina*	225.2	**6.54**	**7.40**
*Euclea natalensis*	214.3	**5.30**	**5.00**
*Helichrysum aureonitens*	305.2	**8.25**	11.40
*Heteropyxis natalensis*	259.7	NT	NT
*Lobostemon fruticosus*	313.4	NT	NT
*Moringa oleifera*	271.6	NT	NT
*Peltophorum africanum*	332.5	NT	NT
*Prunus africana*	249.1	NT	NT
*Ricinus communis*	82.0	NT	NT
*Senna petersiana*	328.7	NT	NT
*Sutherlandia frutescens*	301.9	32.20	42.30
*Terminalia sericea*	232.2	NT	NT
*Ziziphus mucronata*	271.6	NT	NT

LC_50_: lethal concentration of extracts to 50% of Vero cells; EC_50_: concentration of extracts that scavenge 50% of radicals; NT: not tested. Positive control, doxorubicin LC_50_ = 10 µg/mL, ascorbic acid EC_50_ = 2.50 and 2.30 µg/mL for 2,2-diphenyl-1-picrylhydrazyl (DPPH) and 2,2′-Azino-bis(3-ethylbenzothiazoline-6-sulfonic acid) diammonium salt (ABTS) assays, respectively. Bold values indicate extracts, which are considered potent.

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
