# Peer review of "In Vitro Evaluation of Anti-Rift Valley Fever Virus, Antioxidant and Anti-Inflammatory Activity of South African Medicinal Plant Extracts"

_viruses, 2021, doi:10.3390/v13020221_

Round 1

Reviewer 1 Report

The authors have addressed the comments raised by this reviewer. There are a few minor comments that should be addressed:

I would suggest removing the last sentence from the abstract (should be mentioned in the discussion only).

line 95: add reference(s)

The authors have nicely expanded the introduction by adding additional background information. However, I feel that the information presented in lines 101 to 118 could be moved to the discussion.

lines 294 to 297: add reference(s)

Reviewer 2 Report

The authors have re-written some parts of the manuscript and added some information and the paper now  is certainly improved.

However I don’t find any change regarding my concerns in the previous report, the figures are the same so I guess the experimental part has not changed. As I said, when following the procedure as described, the authors are detecting some antiviral activity ON THE VIRUS PARTICLE but not during the infection process in the cells; actually I am happy to find a new paragraph included in the discussion (lines 331-336) about this point. But I think that this does not prove that the protection from oxidative responses provided by the plant extracts (as shown in figure 4) is connected to the antiviral activity claimed. As I understand, the authors show that 1- the incubation of the virus with (some of the) plant extracts may change the morphology of viral particles thus impairing their infectivity (figure 2); and 2- (some of the) plant extracts do exert anti-oxidative properties on cells, but no assay shows that this leads indeed to antiviral protection in cells (I think a way to do that would be to analyze how viral infection proceeds on Vero cells after treatment with those samples shown in figure 4).  Actually both results can be valid and interesting but I think this should be more clearly stated in the paper, because I find that the current redaction is somehow focused on bring both results together (final lines in the discussion) so the readers may reach the wrong conclusion .

Further questions:

Please check the correct reference numbering  (I think nr 7,8,9 in lines 53-58) are wrong, probably due to new ones.

Line 412- I think “cell wall proteins”  is not correct: as far as I know, mammalian cells have membrane, not wall.

Line 462 - it is still confusing: “1x104.8 viral titer/ml”- I guess this “viral titer” means TCID50 (Tissue culture infectious dose) . Please correct it.

Author Response

This manuscript is a resubmission of an earlier submission. The following is a list of the peer review reports and author responses from that submission.

Round 1

Reviewer 1 Report

This is a very interesting paper in which the authors evaluate antiviral activities of 20 medicinal plant extracts against Rift Valley Fever virus. The text is very well written and organized, and the data is presented and analyzed in a systematic fashion.

Comment 1. The main area where improvement is needed is in describing a few details of the methodology, for example, when and how long you add the extracts and virus to the cells (see below).

Results 2.2 - You need to describe how you added the extracts and virus to the cells. Were they added simultaneously? Was the drug added an hour before infection? etc.

Figure 3 - please indicate on the graph (maybe with arrows?) at which times you added the extracts and virus.

Results 2.6 - how long were the cells pre-treated with the extracts?

Comment 2. A few acronyms also need to be defined for readers that may not be familiar with them (listed below).

Line 16 - define RVFV
Line 16 - define CPE
Line 19 - define ROS/RNS

Table 1 - please define DPPH and ABTSs somewhere in the text, for example, under "2.1 cytotoxicity results". Or perhaps move Table 1 to later in the text when you talk about these assays.

Results 2.3 - define DPPH; what timepoint are you using to measure this data?

Results 2.4 - define ABTS; what timepoint are you using to measure this data?

Comment 3. While the extracts clearly have an impact on ROS/RNS reduction, it is not clearly discussed that the extracts may have other impacts on RVFV infection and these limitations should be mentioned in the discussion. Incubating the virus stock with the extract before addition to the cells, for example, can tell you if the compound affects virus structure. Adding the compound at various times after virus infection will also provide insight into which step(s) of viral replication may be blocked by the compound. some compounds can block multiple steps of viral replication.

minor changes:

Table 1 - Why were all compounds not tested for DPPH and ABTS?

Results 2.1 - It would be helpful to state that you chose the threshold 10^5 TCID50 because it lowers viral load by 2 logs compared to the virus control

Results 2.2 - what time point are you using to measure TCID50 in Figure 2?

Line 175 - add the missing bracket ")".

In the discussion: why did you use these three different tests - DPPH, ABTS and LPS-induced ROS? How do they compare to each other and what are the differences between them? Do they measure the outcome of different signaling pathways or should they all have similar results?

In the introduction: are there any in vivo studies that show that limiting ROS/RNS can reduce viral load or pathology? what role does ROS/RNS have in virus replication and what effect should limiting it have an the overall outcome of infection in cells or animals/people?

Reviewer 2 Report

The paper describes a study of plant extracts displaying some activity as inhibitors of oxidative responses as potential antiviral substances able to impair RVFV growth. The subject is interesting because of the importance of the disease, Rift Valley fever, and because of the potential use of plant extracts both by economic and social reasons as already marked by the authors. Out of 20 initial extracts 8 are subjected to a more detailed analysis.

Even though the biochemical analysis focused to measure activities linked to oxidative stress (results shown in sections 2.3, 2.4, 2.5 and 2.6) give positive and interesting results, I do not think the virological studies are correctly designed. In my opinion the approach to measure antiviral activity is wrong:  according to M&M, virus (a fixed amount- confusing “100 viral titer/ml”, see further) and extracts (at a concentration 100 mg/ml) are mixed and incubated overnight and then 10-fold serially diluted and mixed with cells- so the extract is expected to act during this incubation mainly ON THE VIRUS PARTICLE but not during the infection process in the cells???  Following this procedure the active substance inhibiting the virus would also be diluted, and the underlying idea that its inhibitory activity has something to do with protecting from oxidative responses produced during infection is not easy to understand. In my opinion the correct assay would be to mix a fixed amount of virus with diluted doses of the extract and/or pretreatment of the cells with diluted doses of the extract and then proceed with the viral infection. Thus I think that the main point of the work is not valid in the way it is presented.

Other questions:

What does it mean “100 viral titer/ml” (line 325)? If 100 is not a typing mistake the dilution of 1 to 8 log units has no sense.

I never heard about the RTCA assay so I find the RTCA results difficult to interpret– the text says “Decreased CI values” (line 156) for infected cells  but for me the corresponding blue line in figure 3 shows increased values. Besides, based on my experience, I find it strange to see some effect in infected cells only after 90 h,  i.e. close to 4 days post infection. A very low amount of virus could explain what I find a very slow infection, but this data is missed in the M&M section.  

Some suggestions:

The Abstract is confusing, the main idea should be stated in a more direct and clear way. Besides, it would be more interesting for readers to include a quantitative value of the viral reduction achieved.

Check “WHO AFRO - Outbreaks and Emergencies Bulletin” for possible updating on the very last outbreaks (Lines 48-50).

I would find it useful to see – maybe as supplementary information- the curves for the analysis and calculation of the LC50 , at least for the 8 selected extracts.

I would find it useful to include a new column in Table 1 showing the individual MNTC (maximal non-cytotoxic concentrations). Maybe I misunderstood, but it is strange that in spite of the different values of LC50  (between 82-400 mg/ml)  the assay of antiviral activity is done in all cases at 100 mg/ml- which is over the lethal dose for one of the extracts (R.communis, 82.0) or close to it (A. afra,  151.8).

Regarding Figure 4, it could be interesting to do these measures in RVFV-infected cells, in order to connect the phenomena under study (infection-scavenging activities). In the absence of such an assay the questions presented in the paper (1- scavenger activity of the extracts and 2- antiviral activity) are unrelated and any discussion on the mechanisms is speculative.

Lines 218-222- at the end of section 2.6 are too speculative for a Results paragraph.

The sentence in line 334 “ No drug control was used in this study since there are no antiviral drugs approved for the treatment of RVFV infections” must be nuanced;  no anti-RVFV-drug is approved to treat the infection but a number of antiviral molecules have been successfully tested in in vitro and in vivo assays.

Typing mistakes: Line 342: 104; line 396: conclussion

Reviewer 3 Report

The manuscript entitled ‘‘In-vitro anti-Rift Valley Fever Virus activity, reduction of free radicals and suppression of LPS-induced ROS/RNS production’’ (Garland K. More et al.) described the antiviral effect of twenty plant extracts against RVFV in Vero cells.

The authors assayed the ability of these extracts to neutralize the virus by neutralization CPE assay. Furthermore, they study their free radicals scavenging potency of those extracts with best antiviral activity measuring the reduction of free radicals and the LPS-induced ROS and NOS levels in Vero cells.

The approach of the study is the correct. The experiments carried out to demonstrate the antiviral and scavenging activity have been well designed. The results obtained are solid and clarifying and the discussion is well written.

Major comments:

 It will be helpful if authors could explain why they use only one cell line in the study and they don’t use others like human liver cancer cell line HepG2 closer to a real infection.

Minor comments:

Authors should check at least the capital letters of RVFV virus, page 2 lines 54-55.

Please authors should review page 6 line 175, the parentheses is missed.

Lines 313 and 342, please replace "all the wells" or rewrite the sentences.

Line 376 add the superscript.

Line 396, correct conclusion.

Reviewer 4 Report

This study by More and colleagues describes the evaluation of extracts from twenty medicinal plants known to have antiviral properties for their specific antiviral activity against Rift Valley fever virus. RVFV is an arbovirus with the ability to cause severe disease (including hemorrhagic fever and encephalitis) in humans. In livestock, RVFV infection can result in large numbers of abortions and can have a tremendous economic effect.  Until now, no approved treatment options are available. It has previously been shown that RVFV infection is associated with increased ROS/RNS secretion, which can lead to cell death and DNA damage. The goal of this study was to screen aqueous-methanolic crude extracts of 20 medicinal plants for radical scavenging activity in vitro.

The authors performed a series of assays to characterize cytotoxicity and identified that eight plant extracts that showed activity in reducing RVFV titers. Furthermore, they demonstrate that these plant extracts have free radical scavenging activity and suppress LPS-induced ROS/RNS in vitro. Overall the study is interesting and demonstrates that medicinal plant extracts should be further characterized for their antiviral properties. The following comments should be addressed:

In Figure 2, the y-axis should indicate that the TCID50 is shown. It would be interesting to also show at least the dose-response curves for the selected eight plant extracts with stronger antiviral activity. The statement made in line 333 that no control drugs were included because none are approved is not valid. Multiple compounds with antiviral activity against RVFV have been identified (and referenced by the authors). At least one of these should have been included to demonstrate that the antiviral assay was working.

Figure 1 is not contributing to the study and should be removed. If the authors would have performed assays to determine the expression levels of RVFV proteins, the figure would be helpful, but not in the context of this study.

Also, the title should be changed and be more specific (e.g., include antiviral activity from medicinal plant extracts).

Please verify that all references are listed. There were several missing (e.g., line 85 references for HSV-1, influenza and RVFV are missing; line 96 Keck et al 2015 for bortezomib and Brahms et al 2017 for sorafenib).

Line 143: How was the threshold of 10^5 TCID50 chosen?

Line 396: Please correct spelling of “Conclusion”

In the Material and Methods, no information is provided regarding the biosafety level at which RVFV has been handled.